# Interpreting Operation Selection in Differentiable Architecture Search: A Perspective from Influence-Directed Explanations

**Miao Zhang**[1,2], **Wei Huang**[3], **Bin Yang**[4]*
[1]Harbin Institute of Technology (Shenzhen)   [2]Aalborg University
[3]RIKEN AIP   [4]East China Normal University
zhangmiao@hit.edu.cn; wei.huang.vr@riken.jp; byang@dase.ecnu.edu.cn

## Abstract

The *Differentiable ARchiTecture Search* (DARTS) has dominated the neural architecture search community due to its search efficiency and simplicity. DARTS leverages continuous relaxation to convert the intractable operation selection problem into a continuous magnitude optimization problem which can be easily handled with gradient-descent, while it poses an additional challenge in measuring the operation importance or selecting an architecture from the optimized magnitudes. The vanilla DARTS assumes the optimized magnitudes reflect the importance of operations, while more recent works find this naive assumption leads to poor generalization and is without any theoretical guarantees. In this work, we leverage influence functions, the functional derivatives of the loss function, to theoretically reveal the operation selection part in DARTS and estimate the candidate operation importance by approximating its influence on the supernet with Taylor expansions. We show the operation strength is not only related to the magnitude but also second-order information, leading to a fundamentally new criterion for operation selection in DARTS, named ***Influential Magnitude***. Empirical studies across different tasks on several spaces show that vanilla DARTS and its variants can avoid most failures by leveraging the proposed theory-driven operation selection criterion.

## 1 Introduction

Neural Architecture Search (NAS) has been successfully applied to automating the process of neural network design for a broad of deep learning fields [35, 41, 48, 50]. However, the early NAS methods are often heavily computational-expensive and require hundreds or even thousands GPU days [34]. To improve the efficiency for more practical applications, a lot of recent works are proposed based on one-shot (also known as weight-sharing) paradigm [33] to improve the search efficiency significantly. *Differentiable ARchiTecture Search* (DARTS) [27], as the most popular one-shot NAS method, has dominated this area since its appearance. DARTS leverages continuous relaxation to convert the intractable operation selection problem into a continuous magnitude optimization problem, which can be gracefully solved by gradient-descent based on the bi-level optimization [8]. A discrete architecture is then derived based on the optimized magnitudes through operation selection.

Although the continuous relaxation makes DARTS able to optimize the magnitudes with rigorous theoretical formulation and guarantees [51] in the supernet training part, it poses an additional challenge in selecting a discrete architecture from the optimized magnitudes that there is no existing work can give a theoretical explanation. DARTS is based on the weight-sharing paradigm [33], which

---

*Corresponding Author.

36th Conference on Neural Information Processing Systems (NeurIPS 2022).

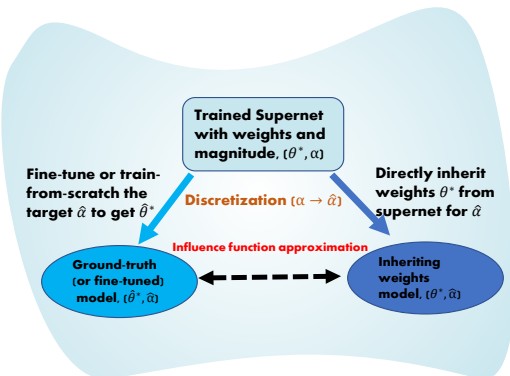

Figure 1: **Pictorial depiction of discretization part in DARTS.** There is a gap between the train-from-the-scratch (or fine-tuned) weights $\hat{\theta}^*$ based on $\hat{\alpha}$ after discretization and the trained supernet weights $\theta^*$. This paper leverages influence functions to quantify the disparity between $\mathcal{L}(\hat{\theta}^*, \hat{\alpha})$ and $\mathcal{L}(\theta^*, \hat{\alpha})$ and show how weights $\theta$ changes with $\alpha$. The discretization process can be the *argmax* adopted by DARTS [27], the perturbation-based DARTS-PT [40], and so on.

assumes that the derived discrete architecture with the trained supernet weights can predict its train-from-scratch performance. However, there is a gap between the two groups of weights, as depicted in Figure 1. The vanilla DARTS assumes the optimized magnitudes can reflect the importance of the candidate operations, while Wang et al. [40] empirically show that the operation magnitude does not necessarily indicate how important the operation is in a trained supernet. For example, DARTS prefers *skip-connection* [45] as its magnitude always dominates the remaining operations, while an architecture with intensive *skip-connections* generally leads to poor performance. Accordingly, Wang et al. [40] proposed a heuristic perturbation-based method, DARTS-PT, to measure each candidate operation's influence on the supernet by alternatively removing each one to calculate the supernet performance drop. Despite DARTS-PT can partially relieve the failure of DARTS, we empirically found that DARTS-PT still prefers *skip-connection* than other operations. One potential reason is that *skip-connection*'s magnitude is much higher than other candidate operations, and directly removing *skip-connections* usually greatly deteriorates the supernet performance. In addition, DARTS-PT lacks theoretical explanations and guarantees. For example, the intuitive way to show the importance of a candidate operation importance is the leave-one-out retraining [22], while which should fine-tune the supernet to get the performance drop after removing every operation rather than directly measuring the performance drop as DARTS-PT, since there also exists a gap between the two groups of weights, as visualized in Figure 1. Section 3 also give an analysis on why perturbation-based method fails.

This paper focuses on filling the gap of theoretical explanations for the discrete operation selection part in DARTS. Instead of considering a heuristic approach to measure the operation importance [27, 40], this paper first rigorously shows how the architecture parameters discretization poses an influence on the supernet weights in DARTS, and estimates the supernet performance drop accordingly. Our method is motivated by influence function [22], which is originally designed to understand the effect of removing an individual training point, while we leverage Taylor's approximation to interpret the supernet performance drop when removing a candidate operation. In this way, we show the operation strength is not only related to the magnitude but also second-order information, and we accordingly design a new criterion for the operation selection in DARTS, called ***Influential Magnitude*** ($\mathcal{I}_\mathcal{M}$). Then, to avoid calculating the inverse of the Hessian matrix with respect to the model weights, we further utilize the Neumann series [29] and Sherman-Morrison formula [37] to approximate the Inverse-Hessian-Vector Products (IHVPs), making the proposed method applicable to DARTS. Finally, we verify the effectiveness of our method on several NAS benchmark datasets and the common DARTS search space. To summarize, we make the following contributions:

- This paper deepens our understanding of the operation selection in DARTS. We reformulate the operation selection in DARTS by approximating its influence on the supernet with Taylor expansions, which can thus gracefully interpret how the validation performance changes when selecting different operations without any additional fine-tuning.

- We theoretically reveal the operation strength is not only related to the magnitude but also the second-order information, and accordingly derive a fundamentally new criterion to measure the operation sensitivity, which we call **Influential Magnitude** ($\mathcal{I}_\mathcal{M}$), for architecture selection in DARTS. Further, to make the proposed criterion practical, we devise several methods to estimate the Inverse-Hessian-Vector Products (IHVPs) in calculating the second-order information.

- Extensive experiments verify the effectiveness of the proposed criterion, which significantly improves the performance of vanilla DARTS and other baselines on the NAS benchmark datasets and the common DARTS search space. We show DARTS is still a strong NAS baseline when considering a more reasonable and theory-driven operation strength metric.

## 2  Preliminaries: DARTS and Influence Function

Typically, NAS focuses on finding a cell structure that is represented as a directed acyclic graph (DAG) with $N$ nodes and $E$ edges affiliated with operations, which is a discrete and hard-to-optimize problem. To enable the gradient-descent for architecture search, **DARTS** [27] leverages *softmax* as the continuous relaxation function to transform the architecture search into a continuous magnitude optimization problem:

$$\alpha_o = \frac{\exp(\bar{\alpha}_o)}{\sum_{o' \in \mathcal{O}} \exp(\bar{\alpha}_{o'})},$$

where $\mathcal{O}$ contains all candidate operations and DARTS is to optimize the magnitude $\alpha_o$, which can be formulated as a bi-level optimization problem [8]:

$$
\begin{aligned}
\min_{\alpha} \quad & \mathcal{L}_{val}(\theta^*(\alpha), \alpha) \\
\text{s.t.} \quad & \theta^*(\alpha) = \mathrm{argmin}_\theta \, \mathcal{L}_{train}(\theta, \alpha),
\end{aligned}
\tag{1}
$$

where $\alpha$ is the continuous architecture representation and $\theta$ is the supernet weights. The nested formulation in DARTS is the same as the gradient-based hyperparameter optimization with bi-level optimization, which has been validated in a broad applications [13, 31, 32]. After the bi-level optimization, DARTS usually considers heuristic methods to derive the final architecture, where the most popular one is to select the operations with the highest magnitudes, $\hat{\alpha} = \boldsymbol{argmax}(\alpha)$. Despite its simplicity, more recent works [7, 40, 45, 47–49, 52] found that DARTS could hardly find satisfactory architectures.

Different from the majority of existing works that attribute the failure of DARTS to its magnitude optimization part, which is under a rigorous theoretical formulation [51], we revisit DARTS from the perspective of the architecture selection and try to reveal this part with theoretical explanations, where we aim to identify the operation that will contribute most to the supernet performance, through the lens of influence functions [22] to mathematically model the architecture selection process instead of heuristics.

**The influence function** [18] is a classic technique from robust statistics that reveals how model parameters change as we upweight or perturb a specific training sample, which has been applied in explaining a lot of modern machine learning applications [2, 22, 30, 53]. More specifically, given a model with parameters $\theta^*$ after trained on a full training set $\mathcal{D}$, influence function aims to study the change of parameters $\hat{\theta}^* - \theta^*$ when retraining the model parameters to $\hat{\theta}^*$ after deleting a specific training point $z$, that $\hat{\mathcal{D}} = \mathcal{D} - z$. Rather than retraining the model for every removed point, Koh and Liang [22] estimated the influence by upweighting the data point with some small $\epsilon$. With leveraging the Taylor expansions, the influence on the model parameters can be estimated as:

$$\mathcal{I}(z, \theta) = \frac{d\hat{\theta}^*}{d\epsilon} = -H_{\theta^*}^{-1} \nabla_\theta \mathcal{L}(z, \theta^*), \tag{2}$$

and its influence on the loss function of a test point $z_{\text{test}}$ is accordingly calculated as:

$$\mathcal{I}(z, \mathcal{L}) = \frac{d\mathcal{L}(z_{\text{test}}, \hat{\theta}^*)}{d\epsilon} = -\nabla_\theta \mathcal{L}(z_{\text{test}}, \theta^*)^\top H_{\theta^*}^{-1} \nabla_\theta \mathcal{L}(z, \theta^*). \tag{3}$$

The detailed derivation of the two influence functions can be found in [22]. Based on Eq.(2) and (3), we can now approximate the validation performance drop when removing a training point $z$ from the training set without conducting the time-consuming retraining.

The original work of influence functions for modern machine learning [22], which is also called as first-order influence functions, only considers a very small change on $\theta^*$ as it only changes one training point, so the first-order Taylor expansion is satisfied to calculate the influence. However, upweighting a group of data points may lead to a large change on $\theta^*$ [4, 5], which violates the small perturbation assumption of the first-order influence functions. To make the the influence function capture the model parameter change more precisely, Basu et al. [4] extended influence functions using second-order approximations under the assumption that the third-order derivatives of the loss function at optimum is small. For the quadratic loss, the third-order derivatives of the loss are zero or very small, and Basu et al. [4] empirically verified that the this assumption also approximately held for the classification problem with cross-entropy loss function.

Different from existing works that analyze the effects on model parameters of removing data points, this paper creatively adapts influence functions to estimate the importance of candidate operations on a trained supernet in differentiable architecture search. We also follow the assumption in [4] that neglect the third-order and higher derivatives of the loss, and analyze the operation selection part, which is an additional process after bi-level supernet training in DARTS, through the lens of influence function in the following sections.

## 3 Interpret Operation Selection through Influence Functions

This section aims to understand how the model parameters $\theta$ (a.k.a. the supernet weights in DARTS) would change when we derive the discrete architecture from the optimized continuous operation magnitudes, a.k.a. architecture selection. In this paper, our analysis only focuses on this architecture selection part after the bi-level supernet training process is finished. To be more clearly, we first define the optimized operation magnitudes by DARTS as $\alpha$ with the optimized supernet weights $\theta^*$, and the validation loss is $\mathcal{L}(\theta^*, \alpha)$ for simplicity. Then, we define the architecture parameters change to $\hat{\alpha}$ after discretization, and the corresponding ground-truth supernet weights and validation loss are defined as $\hat{\theta}^*$ and $\mathcal{L}(\hat{\theta}^*, \hat{\alpha})$, respectively. Rather than deleting a single data point that only brings small changes on the model parameters [22], the change of architecture parameters can lead to a considerable change on $\theta$, so we leverage the second-order approximation [4] to reveal the supernet weights change. With second-order Taylor expansion on $\hat{\theta}^*$ for $\mathcal{L}(\hat{\theta}^*, \hat{\alpha})$, we have:

$$\Delta\mathcal{L} = \mathcal{L}(\hat{\theta}^*, \hat{\alpha}) - \mathcal{L}(\theta^*, \alpha) \approx \mathcal{L}(\theta^*, \hat{\alpha}) + \Delta\theta^T \frac{\partial \mathcal{L}(\theta^*, \hat{\alpha})}{\partial \theta} + 1/2\Delta\theta^T \frac{\partial^2 \mathcal{L}(\theta^*, \hat{\alpha})}{\partial\theta\partial\theta}\Delta\theta - \mathcal{L}(\theta^*, \alpha), \quad (4)$$

where $\Delta\theta = \hat{\theta}^* - \theta^*$, and $\mathcal{L}$ is the validation loss function in Eq.(1). For example, when we only remove one candidate operation as DARTS-PT [40] to measure the operation importance, the influence of this discretization brings to $\theta$ is $\Delta\theta$, and the true validation loss change should be $\mathcal{L}(\hat{\theta}^*, \hat{\alpha}) - \mathcal{L}(\theta^*, \alpha)$ as stated in Eq.(4) rather than $\mathcal{L}(\theta^*, \hat{\alpha}) - \mathcal{L}(\theta^*, \alpha)$ used by DARTS-PT. As shown in Eq.(4), we need to calculate $\Delta\theta$ to get $\Delta\mathcal{L}$ after we change the $\alpha$ to $\hat{\alpha}$, while we try to avoid the time-consuming retraining process through leveraging influence functions [22] to estimating $\hat{\theta}^*$. Based on the implicit function theorem [29, 51], the validation loss change after discretizing $\alpha$ to $\hat{\alpha}$ can be approximated as the following theorem.

**Theorem 1** *Suppose that DARTS obtains the optimized architecture parameter $\alpha$ with supernet weights $\theta^*$ after supernet training, $\alpha$ changes to $\hat{\alpha}$ when conducting architecture discretization, and the train-from-scratch validation loss for $\hat{\alpha}$ is $\mathcal{L}(\hat{\theta}^*, \hat{\alpha})$. If the third and higher derivatives of the loss function $\mathcal{L}$ at optimum is zero or sufficiently small [4], and with $\frac{\partial \mathcal{L}(\hat{\theta}^*, \hat{\alpha})}{\partial \theta} = 0$, we have*

$$\Delta\mathcal{L} = \mathcal{L}(\hat{\theta}^*, \hat{\alpha}) - \mathcal{L}(\theta^*, \alpha) \approx \mathcal{L}(\theta^*, \hat{\alpha}) - \mathcal{L}(\theta^*, \alpha) - 1/2 \frac{\partial \mathcal{L}(\theta^*, \hat{\alpha})}{\partial \theta}^T \frac{\partial^2 \mathcal{L}(\theta^*, \hat{\alpha})}{\partial\theta\partial\theta}^{-1} \frac{\partial \mathcal{L}(\theta^*, \hat{\alpha})}{\partial \theta}. \quad (5)$$

Theorem 1 states that we can estimate the train-from-scratch validation performance of $\hat{\alpha}$ without any retraining. During the discretization process, we can apply the ***argmax*** on all edges by one-shot that $\hat{\alpha} = \textbf{\textit{argmax}}(\alpha)$, or only discretize one edge in each step, or remove one candidate operation from one edge as DARTS-PT [40] which can be formulated as the first part of Eq.(5), $\mathcal{L}(\theta^*, \hat{\alpha}) - \mathcal{L}(\theta^*, \alpha)$. We prefer an iterative discretization process that only applies a small change on $\alpha$ as DARTS-PT, since our theoretical analysis in the next corollary will show that the error bound of our approximation is related to $\Delta\alpha = \hat{\alpha} - \alpha$. Before our analysis, we give the following common assumptions, which are also considered in several recent bi-level optimization problems [9, 15–17, 51].

**Assumption 1** *For any $\theta$ and $\alpha$, $\mathcal{L}(\cdot, \alpha)$ and $\mathcal{L}(\theta, \cdot)$ are Lipschitz continuous with $C_f > 0$ and $C_L > 0$, respectively.*

**Assumption 2** *$\mathcal{L}(\theta, \alpha)$ is twice differentiable with constant $C_H$ and is $\lambda$-strongly convex with $\theta$ around $\theta^*(\alpha)$.*

**Assumption 3** *$\left\| \nabla^2_{\theta \alpha} \mathcal{L} \right\|$ is bounded with constant $C_a > 0$.*

**Corollary 1** *Based on the Assumption 1-3, we could bound the error between the approximated validation loss $\mathcal{L}(\hat{\theta}^*, \hat{\alpha}) = \Delta \mathcal{L} + \mathcal{L}(\theta^*, \alpha)$ and the ground-truth $\tilde{\mathcal{L}}(\hat{\theta}^*, \hat{\alpha})$ in DARTS with $E = \left\| \mathcal{L}(\hat{\theta}^*, \hat{\alpha}) - \tilde{\mathcal{L}}(\hat{\theta}^*, \hat{\alpha}) \right\| \leqslant \frac{K^3}{6} \max \left| \frac{\partial \mathcal{L}^3}{\partial \theta^3} \right|$, where $K = \frac{C_L}{\lambda} \|\Delta \alpha\| + \frac{C_H C_a^2}{2\sigma_{min}^2 \lambda} \|\Delta \alpha\|^2 + o(\|\Delta \alpha\|^2)$.*

Compared with the *argmax* used in DARTS, we prefer the perturbation-based discretization method [40] as which only remove one operation at each step, since the above Corollary 1 show that error bound of the approximated validation loss $\mathcal{L}(\hat{\theta}^*, \hat{\alpha})$ increase with the magnitude of the change of $\alpha$.

## 4 Influential Magnitude for Operation Selection in DARTS

Section 3 provides an iterative solution to measure the operation importance based Eq. (5) through individually removing each operation, while which needs to repeat $n$ times (the number of all candidate operations for all edges in the supernet work). More important, since removing one operation poses a considerable change on $\alpha$ ($\alpha_i \to 0$), the approximation error produced by Eq.(5) is non-negligible which may affects the accuracy as stated by **Corollary** 1. So, a more practical solution to illustrate the importance of each operation is to estimate how the validation performance will change after posing an infinitesimal change on $\alpha$, a.k.a. operation sensitivity. In this section, we aim to investigate the operation importance through the lens of sensitivity [24]. Different from Section 3 that removes one operation ($\alpha_i \to 0$), this section only changes the $\alpha_i$ with an infinitesimal $\varepsilon_i$, which means that we can now conduct first-order Taylor expansion on $\alpha$ for $\mathcal{L}(\hat{\theta}^*, \hat{\alpha})$. Further with second-order Taylor expansion on $\theta$ and assuming the local optimal $\frac{\partial \mathcal{L}(\theta^*, \alpha)}{\partial \theta} = 0$, we have:

$$\Delta \mathcal{L} = \mathcal{L}(\hat{\theta}^*, \hat{\alpha}) - \mathcal{L}(\theta^*, \alpha) \approx 1/2 \Delta \theta^T \frac{\partial^2 \mathcal{L}(\theta^*, \alpha)}{\partial \theta \partial \theta} \Delta \theta + \Delta \alpha^T \frac{\partial \mathcal{L}(\hat{\theta}^*, \alpha)}{\partial \alpha}. \tag{6}$$

Similar to Eq.(4), it is intractable to directly calculate $\Delta \theta$, while we can leverage implicit function theorem to approximate it. Theorem 2 presents how the validation performance changes after we pose an infinitesimal change on $\alpha$.

**Theorem 2** *Suppose that DARTS obtains the optimized architecture parameter $\alpha$ with supernet weights $\theta^*$ after supernet training, and we pose an infinitesimal change on $\alpha$. Based on implicit function theorem and under the assumption that the third and higher derivatives of the loss function at optimum is zero or sufficiently small [4], the change of validation loss can be estimated as:*

$$\Delta \mathcal{L} = \mathcal{L}(\hat{\theta}^*, \hat{\alpha}) - \mathcal{L}(\theta^*, \alpha) \approx -1/2 \Delta \alpha^T \frac{\partial^2 \mathcal{L}(\theta^*, \alpha)}{\partial \alpha \partial \theta} H^{-1} \frac{\partial^2 \mathcal{L}(\theta^*, \alpha)}{\partial \theta \partial \alpha} * \Delta \alpha, \tag{7}$$

*where $H = \frac{\partial^2 \mathcal{L}(\theta^*, \alpha)}{\partial \theta \partial \theta}$ is the Hessian matrix.*

From Theorem 2, we can observe the relationship between $\Delta \mathcal{L}$ and $\Delta \alpha$. To be more specifically, when we only consider an infinitesimal change on $\alpha$, that $\Delta \alpha = \varepsilon \cdot \mathbf{1}$ where $\mathbf{1}$ is a column vector with all ones and $\varepsilon$ is an infinitesimal scalar, the sensitivity of $\alpha$ can be defined as:

$$\frac{\Delta \mathcal{L}}{\Delta \alpha} = -1/2 \cdot \varepsilon \cdot \mathbf{1}^T \frac{\partial^2 \mathcal{L}(\theta^*, \alpha)}{\partial \alpha \partial \theta} H^{-1} \frac{\partial^2 \mathcal{L}(\theta^*, \alpha)}{\partial \theta \partial \alpha}. \tag{8}$$

Accordingly, we propose an ***Influential Magnitude*** to quantify the operation importance through the lens of sensitivity:

**Definition 1** *Influential Magnitude ($\mathcal{I}_{\mathcal{M}}$): Suppose DARTS obtains the optimized magnitude $\alpha$ with supernet weights $\theta^*$ after supernet training, the operation sensitivity can be defined as $\mathcal{I}_{\mathcal{M}} = -\mathbf{1}^T \frac{\partial^2 \mathcal{L}(\theta^*, \alpha)}{\partial \alpha \partial \theta} H^{-1} \frac{\partial^2 \mathcal{L}(\theta^*, \alpha)}{\partial \theta \partial \alpha}$.*

Definition 1 shows we can replace the magnitude $\alpha$ with the operation sensitivity $\mathcal{I}_{\mathcal{M}}$ in the operation selection for DARTS, whose *i-th* value represents the importance of *i-th* candidate operation.

## 5 Practical Calculation on Operation Influence

For a large neural network, it is impractical to calculate the second-order information, e.g., the Hessian matrix $H$, let alone the inverse of Hessian. Generally, the core challenge in calculation of Eq.(5) and Eq.(7) is the Inverse-Hessian Vector Products (IHVPs), $H^{-1}v$ [14, 22, 37]. As to the term of $\frac{\partial^2 \mathcal{L}(\theta^*, \alpha)}{\partial \theta \partial \alpha}$, we follow DARTS [27] to utilize the finite difference approximation for the practical calculation. The first technique for IHVPs is the conjugate gradient (CG) which converts the matrix inverse problem into an optimization problem with $\delta$-optimal solution. However, the standard CG can be slow with a large dataset as it needs to go through the whole training points in each step [1]. The original paper [22] using influence function for modern machine learning problem leverages the Neumann series to approximate the inverse, that $H^{-1} = \sum_{k=0}^{\infty} (I - H)^k$, with only consider the first $t$ terms to approximate $H^{-1}$. However, there is a strict condition in the Neumann series approximation that $\|I - H\| \leq 1$, which can not be guaranteed in the practical implementation. Differently, we consider that $H^{-1} = \gamma(\gamma H)^{-1} = \gamma \sum_{k=0}^{\infty} (I - \gamma H)^k$, where $\gamma$ is a small enough value. The following lemma gives our Neumann series based implementation for IHVPs.

**Lemma 1** *With small enough $\gamma$, and assuming $\mathcal{L}$ is $\lambda$-strongly convex at optimum, $H^{-1}v$ can be formulated as:* $H^{-1}v = \gamma \sum_{k=0}^{K} [I - \gamma H]^k V_0 = V_0 + V_1 + ... + V_K$, *where* $H = \frac{\partial^2 \mathcal{L}(\theta^*, \alpha)}{\partial \theta \partial \theta}$, $V_0 = \gamma v$, *and* $V_1 = \gamma(I - \gamma H)V_0, ..., V_K = \gamma(I - \gamma H)V_{K-1}$.

In the practical implementation, it is still hard to make sure $\mathcal{L}$ is $\lambda$-strongly convex and not easy to select an appropriate $\gamma$. The work [5] empirically showed that the stochastic Neumann series estimation for IHVP is somehow erroneous, especially when the network is deep. As we know, the true Fisher information matrix converges to the Hessian matrix with the training loss approaches to zero. However, it is usually infeasible to calculate the exact Hessian or Fisher matrix in a large neural network. A practical approximation is called as empirical Fisher, which has been verified to be effective in variety of applications [3, 19]. In addition, an important advantage from empirical Fisher is that it allows to approximate the inverse of the Hessian based on the Sherman-Morrison formula. Although the empirical Fisher can not exactly match the true Fisher and there are several active researches on its applicability [23, 38], empirical Fisher is significantly more computationally-efficient, which obtains a good trade-off between the approximation and practical efficiency [37]. Apart from Neumann series, we also consider Sherman-Morrison formula [14, 37] $(A + uv^T)^{-1} = A^{-1} - \frac{A^{-1}uv^T A^{-1}}{1 + v^T A^{-1}u}$ to calculate IHVPs. With empirical Fisher $F = \frac{1}{n} \sum_{j=1}^{n} \nabla_\theta \mathcal{L}_j (\nabla_\theta \mathcal{L}_j)^T$ replacing true Fisher $\hat{F} = \nabla_\theta f (\nabla_\theta f)^T$, IHVPs can be calculated as following.

**Lemma 2** *When assume the empirical Fisher can approximate the Fisher matrix, and $H$ is the Hessian matrix $\frac{\partial^2 \mathcal{L}(\theta^*, \alpha)}{\partial \theta \partial \theta}$ in the optimal point, the IHVPs $H^{-1}v$ can be formulated as:* $H^{-1}v = F_n^{-1}v = F_{n-1}^{-1}v - r_n \frac{r_n^T v}{N + \nabla_\theta \mathcal{L}_n^T r_n} = \eta^{-1}v - \sum_{j=1}^{n} r_j \frac{r_j^T v}{N + \nabla_\theta \mathcal{L}_j^T r_j}$, *where* $\mathcal{L} = \ell + \eta \mathcal{R}(\theta)$ *that $\ell$ is a cross-entropy loss and $\mathcal{R}$ is the regularization term,* $F_n = \frac{1}{n} \sum_{j=1}^{n} \nabla_\theta \mathcal{L}_j \nabla_\theta \mathcal{L}_j^T$ *is the empirical Fisher, and* $r_j = F_{j-1}^{-1} \nabla_\theta \mathcal{L}_j$ *which can be recurrently calculated through* $r_j = \eta^{-1} \nabla_\theta \mathcal{L}_j - \sum_{i=1}^{j-1} r_i \frac{r_i^T \nabla_\theta \mathcal{L}_j}{N + \nabla_\theta \mathcal{L}_i^T r_i}$.

## 6 Experiments

In Section 3 and 4, we have theoretically interpreted the operation selection through individually measuring each operation's influence on supernet, and accordingly devised an ***Influential Magnitude*** ($\mathcal{I}_\mathcal{M}$) to replace the commonly-used magnitude $\alpha$ for architecture selection. In this section, we conduct a series of experiments to analyze whether our proposed method leads to better and more stable results in differentiable architecture search settings. We consider two NAS benchmark dataset search space, NAS-Bench-201 [12] and

Table 1: Test accuracy (%) on NAS-Bench-201.

| Method | CIFAR-10 | CIFAR-100 | ImageNet-16-120 |
|---|---|---|---|
| Random | 86.61±13.46 | 60.83±12.58 | 33.13±9.66 |
| DARTS | 54.30±0.00 | 15.61±0.00 | 16.32±0.00 |
| DARTS-PT | 89.63±0.19 | 62.35±2.14 | 36.51±2.13 |
| DARTS-IF | 91.84±0.84 | 67.94±1.23 | 42.50±3.30 |
| DARTS-IM | **93.61±0.23** | **71.31±0.40** | **44.98±0.36** |

Full comparison with SOTAs is in Appendix D. We default use Sherman-Morrison for DARTS-IM in this experiment.

NAS-Bench-1shot1 [46], and the most popular DARTS search space [27]. We verified the effectiveness of the proposed theory-driven method compared with two heuristic methods, *argmax* and

perturbation-based selection, and show that the proposed ***Influential Magnitude*** can be applied to DARTS and its variants with greatly enhancing the performance.

## 6.1 Reproducible Comparison on Benchmark Datasets

To empirically verify the effectiveness of our method, we first run a series of experiments on the two NAS benchmark datasets, NAS-Bench-201 and NAS-Bench-1shot1. To make our results reproducible, we only replace the selection part in DARTS with $\mathcal{I}_\mathcal{M}$ and keep the remaining part identical.

Table 1 summarizes the statistical comparison results on NAS-Bench-201 with different *random seeds*, where DARTS-IF, as described in Section 3, follows DARTS-PT that individually removes operation to measure the supernet performance drop, while DARTS-IM, as described in Section 4, calculates the operation sensitivity (influential magnitude) to replace the original magnitude. The **Random** baseline is to randomly select architectures without considering $\alpha$ or $\theta$, and DARTS and DARTS-PT leverage the *argmax* and perturbation-based heuristic selection, respectively. As shown, our DARTS-IF and DARTS-IM both significantly outperform the random baseline and the two heuristic baselines, where DARTS-IM achieves near-optimal results, a **94.29**%, **72.67**%, and **45.93**% test accuracy on CIFAR-10, CIFAR-100, and ImageNet, respectively, in a single run (DARTS-IM with Sherman-Morrison approximation and $N = 20$ under *random seed* 0).

From the searched architectures in NAS-Bench-201, we found that DARTS selects *skip-connection* for all edges as its magnitude usually outweighs other candidate operations. The results of DARTS in NAS-Bench-201, which are even poorer than the random baseline, also suggest that the magnitudes could hardly reflect the operation importance. Although DARTS-PT can partially relieve this issue by selecting one or two more convolutional operations, it still prefers *skip-connections* (selects 3 *skip-connections* for 6 edges) where directly removing them still significantly affects the performance.

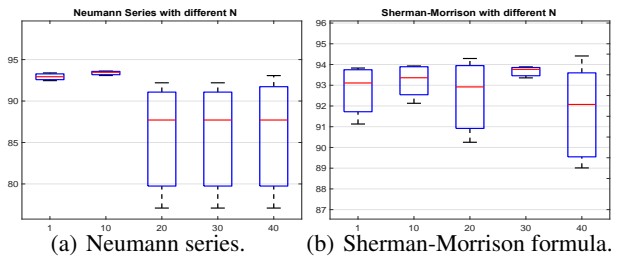

(a) Neumann series.  (b) Sherman-Morrison formula.

Figure 2: Ablation study on $N$ under two approximation methods, where x-axis is N and y-axis represents test accuracy on CIFAR-10. The computational complexity increases with the square of $N$ for Sherman-Morrison formula, and we set the maximal $N = 40$.

On the contrary, our DARTS-IF and DARTS-IM can find valid architectures with more convolutional operations (with only one *skip-connection*). Results in Table 1 verified the effectiveness of our influence functions based selection methods, which outperform baselines by large margins. DARTS-IF is similar to DARTS-PT while with an additional term to approximate the fine-tuned performance and the results that DARTS-IF outperforms DARTS-PT verified the effectiveness of the influence function explanation.

As discussed in Section 5, we can leverage different methods to approximate the IHVPs in our $\mathcal{I}_\mathcal{M}$, and Figure 2 also analyzes different approximation methods along with the number of batches $N$. As demonstrated, increasing $N$ can help the Sherman-Morrison approximation to more accurately catch the operation importance and we empirically find that $N = 30$ is enough in the NAS-Bench-201 dataset. However, we also observe Neumann approximation is not stable which even obtains worse results

Table 2: Best test error (%) on NAS-Bench-1shot1.

| Method | Space1 | Space2 | Space3 |
|---|---|---|---|
| DARTS | 6.17±0.09 | 6.30±0.00 | 6.80±0.00 |
| DARTS-PT | 6.25±0.05 | **6.28±0.06** | 6.69±0.21 |
| DARTS-IM | **6.10±0.24** | 6.53±0.05 | **6.20±0.00** |
| PC-DARTS | 6.37±0.05 | 6.30±0.00 | 6.50±0.00 |
| PC-DARTS-PT | 6.14±0.08 | 6.37±0.12 | 6.38±0.09 |
| PC-DARTS-IM | **5.90±0.24** | **6.20±0.22** | **6.10±0.08** |

with increasing number of batches, showing it is somehow erroneous in estimating IHVPs.

The NAS-Bench-1shot1 is another popular NAS benchmark dataset to analyze differentiable NAS methods, which is built based on the NAS-Bench-101 [44], through dividing all architectures in NAS-Bench-101 into three different unified cell-based search spaces to enable differentiable NAS methods to be directly applied. NAS-Bench-1shot1 contains 6240, 29160, and 363648 architectures in three spaces, respectively, where Space3 is much more complicated than the remaining two spaces.

In NAS-Bench-1shot1, we tracked the searched architecture in every epoch under different operation selection paradigms. We considered two popular supernet training baselines, DARTS and PC-DARTS, and Table 2 summarised the best-searched architectures during the supernet training in different sub-search spaces in NAS-Bench-1shot1. As demonstrated, our $\mathcal{I}_{\mathcal{M}}$ can enhance the performance of DARTS and its variant PC-DARTS in different search spaces of NAS-Bench-1shot1, showing that our influence magnitude (IM) can find more competitive architectures than the default *argmax*, especially in the more complicated space. Compared with the perturbation strategy (PT), our IM is more effective to measure the operation sensitivity with obtaining 5 better results in 6 cases.

## 6.2 Reproducible Comparison on DARTS Search Space

The DARTS search space is the most commonly-used space to evaluate the NAS methods, while most SOTAs only report the performance of their best searched architectures without analyzing the search statistics. To encourage reproducibility, we train the supernet with the same random seeds, and the trained supernet is utilized to derive the architecture under different operation selection strategies. The comparison results are provided in Table 3, from which we can see that, compared with the *argmax* baseline, the architectures searched by our **Influential Magnitude** obtain better performance on CIFAR-10 and ImageNet than DARTS and its variant

Table 3: Search results on DARTS space.

| Method | CIFAR-10 Test Error (%) | | ImageNet |
| | Single | Multi* | Best |
| --- | --- | --- | --- |
| DARTS | 2.76±0.09 | 3.02±0.45 | 26.9 / 8.7 |
| PC-DARTS | 2.57±0.07 | 2.92±0.26 | 25.1 / 7.8 |
| DARTS-PT | **2.61±0.08** | 2.89±0.31 | 26.1 / 8.2 |
| DARTS-IM | **2.50±0.10** | **2.70±0.18** | **25.0 / 7.6** |

* We run the architecture search with multiple times, and average the different derived architecture's test error.

PC-DARTS, and DARTS-PT on ImageNet, implying our method is more stable to find valid and competitive architectures. We also plot the searched architecture examples with different selection methods in the DARTS space in the supplemental material. Although DARTS, PC-DARTS and DARTS-PT can find satisfying architecture after multiple runs, we found the *argmax* is hard to consistently obtain stable results. In addition, different from the perturbation-based operation selection which needs to fine-tune the supernet after the discretization of each edge that costs 0.4 additional GPU day for DARTS, our influential magnitude is calculated by one time which only costs hundred seconds, making our method is much more efficient.

## 6.3 Discussion on the Robustness of Search

The robustness has been a critical concern in the differentiable architecture search community since Zela et al. [45] observed a performance collapse that DARTS tends to yield degenerate architectures, which especially prefers the *skip-connections* with the search progressing. Similar to DARTS-PT, we also focus on the operation selection part to explain the failure of DARTS. Figure 3 presents several architectures by DARTS with our $\mathcal{I}_{\mathcal{M}}$ on different search spaces. Apart from NAS-Bench-201, we also consider several tool search spaces (S1-S4) proposed by [45], which are specifically designed to verify the robustness of differentiable NAS meth-

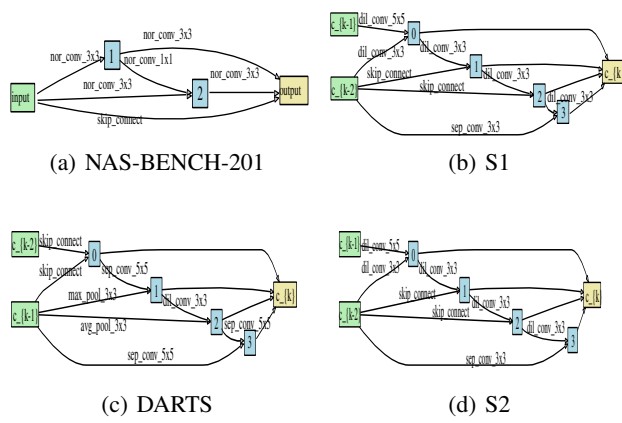

(a) NAS-BENCH-201    (b) S1

(c) DARTS    (d) S2

Figure 3: Searched architectures on different search spaces.

ods. As shown in Figure 3, DARTS-IM can find more meaningful architectures in different spaces. For example, DARTS fails dramatically in most spaces that only selects *skip-connection* for all edges, while DARTS-IM can find architecture with 5 convolutions in NAS-Bench-201, and 6 and 8 convolutions in S1 and S2.

## 6.4 Discussion at Non-Convergence

As discussed in Section 4, our influential magnitude is derived when $\theta$ is in the optimum, with leveraging different methods to approximate the IHVPs. In this subsection, we analyze whether the proposed influential magnitude is still effective to find important operations when $\theta$ is not in the optimum. Figure 4 tracked the performance of searched architecture in each epoch during the search with Sherman-Morrison formula. As shown, our influential magnitude $\mathcal{I}_{\mathcal{M}}$ is still effective when $\theta$ is not in the optimum, while which is getting

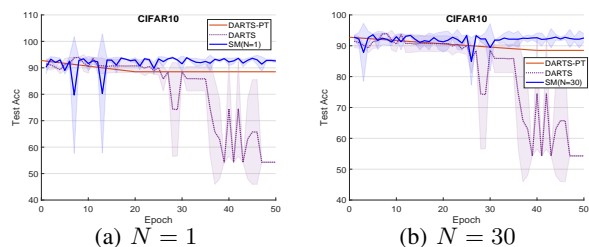

(a) $N = 1$         (b) $N = 30$

Figure 4: Track the architectures during search on NAS-Bench-201 with Sherman-Morrison under different $N$.

more stable with the supernet training. Specifically, as shown in Figure 4 (a) and (b), our $\mathcal{I}_{\mathcal{M}}$ with the different $N$ are both effective, implying that our method is sufficiently informative for the operation selection even under different approximation noises.

More interesting, our DARTS-IM under Sherman-Morrison formula with $N{=}1$ in Figure 4 means that we simply approximate the inverse $H^{-1} = \eta^{-1} \left( I - \eta^{-1} \frac{\nabla_\theta \mathcal{L} \nabla_\theta \mathcal{L}^\top}{1 + \eta^{-1} \nabla_\theta \mathcal{L}^\top \nabla_\theta \mathcal{L}} \right) \propto \hat{H}^{-1}$, that $\hat{H}^{-1}$ is a diagonal matrix that $\hat{H}_{ii} = (\nabla_{\theta_i} \mathcal{L} \nabla_{\theta_i} \mathcal{L})$, which is also called the diagonal approximation of the inverse Hessian. As shown, with roughly approximated second-order information, Eq. (7) also help DARTS-IM to find more important operations compared with existing heuristic methods.

## 6.5 Discussion on the Limitations in Practice

Although our influential magnitude provides a theory-driven and concise solution for the operation selection in DARTS, we consider several assumptions and approximations for the practical implementation. For example, Basu et al. [4, 5] pointed out that the second-order influence values still underestimated the true group influence values in the deep neural networks, while they also observed the correlation marginally improves with the number of parameters in a group and the second-order influence function generally outperforms the first-order. Fortunately, this paper considers the whole parameters in the supernet, which helps the second-order influence function reach its potential.

The approximation of IHVPs requires the conditions that the Hessian matrix is positive definite [29] or the empirical Fisher can replace the true Fisher matrix [14], and the supernet weight $\theta^*$ is a global minimum of the empirical risk, while which in practice, are hard to hold. In the calculation of $v * \frac{\partial^2 \mathcal{L}}{\partial \alpha \partial \theta}$ this paper also follows DARTS that considers the finite difference approximation for practical calculation. These approximations and assumptions may bring errors for the practical calculation, while empirical results verified that our influential magnitude is sufficiently informative for the operation selection in DARTS.

## 7 Conclusion and Future Work

This paper focuses on the operation selection, which is an essential part while with less attention in the differentiable architecture search. Rather than considering heuristics, this paper introduces the influence functions to theoretically reveal the candidate operation's importance through approximating its influence on the supernet. By leveraging the Taylor's approximation, we gracefully interpret how the validation performance changes during the discretization in DARTS and show that the operation strength is not only related to the magnitude but also the second-order information, where an influential magnitude is accordingly devised for the architecture selection. We based our framework on DARTS and extensive experimental results verified the proposed influential magnitude can be practically used in operation selection to avoid failures in differentiable architecture search. Developing more accurate influential magnitude for architecture search or leveraging the training dynamics [20] to reveal the operation influence in the early of training and also extending influential magnitude to other bi-level optimization problems[21, 28, 29], e.g., network pruning [28], are among directions for future work.

## Acknowledgments and Disclosure of Funding

This work was partially supported by Independent Research Fund Denmark under agreements 8022-00246B and 8048-00038B, the VILLUM FONDEN under agreement 34328, and the Innovation Fund Denmark centre, DIREC.

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
