# OpenReview forum: "Interpreting Operation Selection in Differentiable Architecture Search: A Perspective from Influence-Directed Explanations"
_NeurIPS.cc/2022/Conference — NeurIPS 2022 Accept_

### Official Review · Reviewer_8yW8 · 2022-07-06

**Rating:** 9
**Confidence:** 5
**Soundness:** 4 excellent
**Presentation:** 4 excellent
**Contribution:** 4 excellent

**Summary:**

This paper revisits the operation selection part in differentiable architecture search and proposes to introduce the influence function to interpret the operation strength in the architecture selection. The key idea is to approximate the sensitivity of candidate operations in DARTS through influence functions. Although the DARTS has been the mainstream paradigm in NAS, no theoretical analysis has been conducted on its most important part, the operation selection. Different from the DARTS-PT which uses heuristic perturbation-based methods, the authors provide a theoretical and novel operation selection method with justifications and guarantees through influence functions. Apart from existing magnitude-based and perturbation-based operation selection methods, the authors devise a new influential magnitude to measure the operation importance for NAS, which measures the sensitivity of operation with second-order information. Results on most existing NAS search spaces verify the superiority of the proposed method.

**Questions:**

- The proposed influential magnitude is derived when the supernet is trained to convergence. I am curious about its performance at the network initialization. For example, the SNIP [1] for network pruning at initialization also achieves competitive results. Whether the proposed method also shows a similar phenomenon?
- Rather than applying influence functions to the supernet-based NAS, can it be applied to the model-based NAS method?

[1] Snip: Single-shot network pruning based on connection sensitivity. ICLR, 2019

**Limitations:**

The authors discussed the possible failure and limitations of this paper in Sec 6.5. Although the authors cover the limitations of influence functions in deep neural networks, the computational complexity of the influential magnitude is not well discussed. In addition, the proposed influential magnitude is designed only for the supernet-based NAS, and the discussion of its application to the model-based NAS is lacking.

**Strengths And Weaknesses:**

**Strengths:**
- The proposed method is novel and clear, and the paper is well-written and easy to catch up with. The authors focus on an important while under-investigated aspect in DARTS, operation selection, and for the first time introduce influence functions into the NAS community. The authors devise a general approach for architecture selection by measuring the sensitivity of candidate connections in architecture. This paper is the first attempt to adapt the influence functions theory to the network connection rather than the training data points, so as to make it possible to extend to other applications, e.g., network pruning and model quantization.
- The author, for the first time, mathematically formulate the operation selection part in DARTS in terms of the sensitivity, which is completely different from existing magnitude-based (DARTS) and perturbation-based (DARTS-PT) methods. By leveraging Taylor approximation and implicit function theorem, the authors derive a new criterion under several mild assumptions. The theoretical analysis is sound and rigorous which is lacking in this community, and the authors also provide several approaches for practical implementation.
- The authors conducted sufficient and solid experiments in a variety of NAS search spaces, including NAS-Bech-201, 1Shot1, DARTS, and S1-S4, along with a lot of insight ablation study and discussion. I especially appreciate the reproducible results along with the open-sourced code. The results on different search spaces and datasets verify the superiority of the proposed influential magnitude, which is inspiring and impressive with reproducibility. Although improvements on DARTS space are somehow marginal as we understand the results in this search space are usually reported with the best single search run in existing literature, the authors visualize the search architectures which avoid performance collapse that exists in the magnitude-based selection methods, making the results more convincing.
- This work is relevant to the NeurIPS community, especially to the neural architecture search community. In the long run, I could see this paper raises awareness in the community to be conscious about the operation selection part in the supernet-based NAS, and also encourages the investigation to adapt the influence functions theory to other parts of the machine learning community, rather than only the training data points.

----------
**Weaknesses:**
- The author does not provide an algorithm framework to show the general flow of the proposed method, making it hard to get the core idea at the first glance.
- The DARTS-IF appears in Sec 6.1 without any descriptions before. The author is suggested to describe the algorithm flow of DARTS-IF and DARTS-IM.
- The potential directions are not well discussed. For example, this paper adopts the influence functions to measure the operation importance in a supernet, while I think it is inspiring to design an influential magnitude for the structured network pruning in the future which needs more discussion.
- Although this paper nicely presents the operation selection with influential magnitude, some typos also exist.

---

> ### Author Response · Authors · 2022-08-02
> **Response**
>
> Thanks for the constructive comments! We hope our answers below address all your concerns.
>
> 1. The author does not provide an algorithm framework for DARTS-IF and DARTS-IM.
>
> Thanks for the constructive suggestion. We add a new section in Appendix H to describe DARTS-IF and DARTS-IM (which are sketched in Algorithm 1 and Algorithm 2, respectively), and the hyperparameter setting in the two algorithms. Our DARTS-IF shares a similar perturbation paradigm as DARTS-PT, while without any fine-tuning. Rather than using the validation performance drop to indicate the operation importance, DARTS-IF leverages influence functions to predict the loss change. Different from DARTS-IF and DARTS-PT, our DARTS-IM has a similar paradigm as DARTS, while whose operation importance is calculated based on $\mathcal{I_M}$, rather than the optimized $\alpha$. In addition, the operation importance in our DARTS-IM can be calculated by one shot, rather than iteratively going through all candidate operations in a supernet as the perturbation paradigm.
>
> 2. The potential directions are not well discussed, e.g., lacking structured network pruning.
>
> Thanks for the insightful suggestions. We have revised our future work accordingly, with discussing the potential of influence functions based structured pruning.
>
> 3. The performance of influential magnitude in the unconvergence.
>
> Figure 4 draw the performance of our DARTS-IM with the supernet training. Although DARTS-IM can be applied in the initialization or the early training stage, it is not as stable as it performs after supernet training.
>
> 4. Rather than applying influence functions to the supernet-based NAS, can it be applied to the model-based NAS method?
>
> Yes, it can be applied to the model-based NAS method. Although it is impossible to directly introduce influence magnitude for the architecture selection in model-based NAS, we can leverage it to find the important edge or candidate operations to help the heuristic algorithms (evolutionary algorithm or random search) to find better architectures in the model selection.

---

> > ### Comment · Reviewer_8yW8 · 2022-08-09
> > **Thanks for the detailed response**
> >
> > Thanks for addressing my concerns. After reading the comments and response, I think this paper leverage an interesting approach, influence function, to explain the operation selection part in DARTS, with theoretic analysis and comprehensive experiments. Overall, I agree with Reviewer 3tjK and D82N that the paper makes a solid attempt to improve operation selection in DARTS. My overall rating for this paper remains the same.

---

### Official Review · Reviewer_3tjK · 2022-07-10

**Rating:** 7
**Confidence:** 4
**Soundness:** 3 good
**Presentation:** 2 fair
**Contribution:** 3 good

**Summary:**

The paper studies the architecture selection method for differentiable NAS.
It proposes an influential magnitude score based upon the influence function to how sensitive is the supernet loss w.r.t. changes in the continuous relaxed architecture parameter, which can be used to determine the operation strength for discretizing final architecture from DARTS Supernet.
The advantage of the proposed is two folds: 1). Theoretically grounded  2). Practically, require no fine-tuning step compared with the perturbation-based selection method proposed in DARTS-PT.
Empirical results are reported mainly on NAS-Bench-1shot1, NAS-Bench-201, and DARTS Space (CIFAR-10), which show favorable results compared with relevant baselines (i.e. DARTS and DARTS-PT).

**Questions:**

1. In Section 6.1, is the search on NAS-Bench-201 conducted on three datasets separately, or just CIFAR-10?
2. Figure 3 is skewed due to resizing and is hard to read, you may want to modify it.
3. Maybe you can try to use different colors for each curve in Figure 4 to make it easier to match with corresponding legends.

**Limitations:**

Yes, the paper explicitly addresses its limitations.

**Strengths And Weaknesses:**

Strength:

1. Overall, the paper provides a solid attempt to improve architecture selection in NAS. The analysis in Section 3 and 4 on how to derive the influence of changing architecture parameter alpha on the validation loss through model weight is sound, and the corresponding theoretical results are satisfactory.
2. The proposed influence magnitude for measuring operation strength is theoretically justified and intuitive.
3. Empirical evaluations are overall throughout and solid, with an exception which I will explain in the weakness section.

Weakness:

1. The author displays the discovered architectures on S1-S2. It would be better to also report numerical evaluations on S1-S4 in tabular form. The reason is that S1-S4 proposed in RobustDARTS [1] is one of the first discovered scenarios where DARTS fails the most. Including numbers on these benchmarks would help to establish baselines for future works. Another reason for requesting numerical comparison is that architectures with minimal skip connections might not necessarily be optimal.

---

> ### Author Response · Authors · 2022-08-02
> **Response**
>
> Thanks for the constructive comments! We hope our answers below address all your concerns.
>
> 1. Lacking numerical comparison in S1-S4.
>
> Thanks for the constructive comment. We discussed the searched architectures on space S1-S4 in Appendix G and Sec 6.1 (The third paragraph), where our DARTS-IM selects fewer skip connections than DARTS and DARTS-PT, especially on S1 and S2. Following the suggestion, **we also add new comparison experiments in the Appendix to summarize the test errors of different DARTS baselines on S1-S4 in Table 7**, to compare with DARTS, DARTS-ES, and DARTS-PT, where our DARTS-IM achieves the best results in most cases.
>
> 2. In Section 6.1, is the search on NAS-Bench-201 conducted on three datasets separately, or just CIFAR-10?
>
> Yes, we follow the most common setting [1]  to only search on CIFAR10 and report the performance on the three datasets.
>
> 3. Not clear for Figure 3  and Figure 4.
>
> Thanks for the suggestion. We redraw Figure 3 and also use different colors in Figure 4.
>
> [1] NAS-Bench-201: Extending the Scope of Reproducible Neural Architecture Search, ICLR2020

---

> > ### Comment · Reviewer_3tjK · 2022-08-08
> > **Note**
> >
> > I thank the author for the response and added numerical results on S1-S4. The results look mixed compared to DARTS-PT but are substantially better than DARTS/DARTS-ES. My overall rating for this paper remains the same.

---

### Official Review · Reviewer_D82N · 2022-07-10

**Rating:** 6
**Confidence:** 4
**Soundness:** 2 fair
**Presentation:** 2 fair
**Contribution:** 3 good

**Summary:**

The paper improves the operation search method in DARTS by leveraging the proposed influential magnitude. Instead of selecting the architecture operation with operation magnitudes, the influential magnitude uses second-order Taylor terms similar to influence functions. The authors show that their proposed methods improve the final test accuracy on NAS-Bench-201, NAS-Bench-1shot1, and DARTS search spaces and select less number of skip connections, which is the fundamental problem in DARTS-based approaches.

**Questions:**

1. In Section 2, the authors introduce the influence function as the product between the Hessian inverse and the gradient of the loss with respect to its parameters. However, in equation 3, the influence on the test loss function is defined as the product between $s_{\text{stest}}$ (following the notation from the original influence function) and the product between the Hessian inverse and the second-order gradient of the loss with respect to the inputs and model parameters. The second part measures the influence when the data point is perturbed rather than up-weighting the data point, and hence there is a mismatch between equation 2 and equation 3. In summary, in equation 3, $H^{-1} \nabla_x \nabla_\theta \mathcal{L}(z, \theta^*)$ should be replaced with $H^{-1} \nabla_\theta \mathcal{L}(z, \theta^*)$.
2. In equation 6, it would be helpful to state that it is assuming that the gradient of the loss equals 0.
3. Is there a reason why NS performs poorly compared to SH? The core difference is that NS approximates the inverse Hessian, but SH approximates the Hessian with the empirical Fisher. The former is not guaranteed to be PSD while the latter is guaranteed to be PSD, and I wonder if the authors tried using higher damping in NS. The analysis in Appendix F only considers the change in a number of batches.
4. Is there a reason why DARTS-IF performs worse than DARTS-IM in Table 1? I do not understand the statement that “results that DARTS-IF outperforms DARTS-PT verified the effectiveness of the influence function explanation,” as DARTS-IF also uses the influence-based measure, which the iHVP with the gradient of the supernet. I understand that DARTS-IF also required computing the iHVP and was wondering if DARTS-IF is also using SM for iHVP approximation.
5. In Appendix B.3, the authors use finite-difference methods to compute the matrix-vector product, which is a very crude approximation. I believe that this should be described at least in the main text.
6. The results of PT-DARTS in Table 3 seem to be lower than the previously published baseline (which I believe is 2.48). How did the authors obtain results for DARTS-PT (and other baseline methods)?
7. In Section 6.4, the authors aim to show that DARTS-IM works even though the network has not fully converged (the gradient is non-zero). However, I cannot completely understand what Figure 4 is doing. For DARTS-IM, is it doing operation search after x epoch of the supernet training? Why does DARTS fail after 20 epochs of training? Is it due to the skip connection selection issue? Furthermore, I believe that using different colours for DARTS and SM and the comparison with PT-DARTS would be helpful.

I am happy to raise my score if the above issues and questions are addressed.

**Limitations:**

The limitations of the work are described in Section 6.5. I believe the work does not have a potential negative societal impact.

**Strengths And Weaknesses:**

Strengths
* The paper is well written, and the idea of using influence-based operation selection is well motivated.
* The relation to prior work is well discussed, and most derivations in the paper look correct (there is some confusion in some parts of the equation, which I ask in the question section).

Weaknesses
* It is challenging to understand the training scheme of both DARTS-IF and DARTS-IM by reading the paper. I believe that the training algorithm similar to Algorithm 1 in [1] would help understand how both methods are trained and how these two proposed methods differ.
* I find the empirical analysis of DARTS-IF and DARTS-IM to be relatively weak to make the proposed methods convincing. It would be helpful if there were a more extensive analysis compared to DARTS-PT. For example, comparing test losses on search space S1, S2, S3, and S4 on multiple datasets and showing how much improvements DARTS-IM can make compared to PT-DARTS would make the paper much more convincing. Also, quantitative experiments showing that DARTS-IM selecting fewer skip connections would be helpful (instead of qualitative experiments observing some final converged cells). I have additional questions about the experimental procedure below.
* In Section 5, instead of using the Neuman-series to compute the inverse Hessian-vector product, the authors proposed to use empirical Fisher and compute the term using the Sherman-Morrison formula. I believe that the manuscript lacks a proper explanation for this approximation.
* DARTS-IM requires more hyper-parameters to obtain the final architecture as it requires computing iHVP (compared to PT-DARTS). What are additional hyperparameters that need to be tuned compared to DARTS-PT, and how much additional time do you require to tune these parameters? The algorithm of the proposed methods (as described above) could also benefit the readers trying to understand these points.
* While the authors answered yes to the checklist, I could not find a section that describes all training details of DARTS-IF and DARTS-IM.

[1] Wang, R., Cheng, M., Chen, X., Tang, X., & Hsieh, C. J. (2021). Rethinking architecture selection in differentiable NAS. arXiv preprint arXiv:2108.04392.

---

> ### Author Response · Authors · 2022-08-02
> **Response (1/2)**
>
> Thanks for the constructive comments! We hope our answers below address all your concerns.
>
> 1. Lacking training algorithm frameworks and all training details for DARTS-IF and DARTS-IM.
>
> Thanks for the constructive suggestion. We **add a new section in Appendix H to describe DARTS-IF and DARTS-IM (which are sketched in Algorithm 1 and Algorithm 2, respectively), and the hyperparameters setting in these two algorithms**. Our DARTS-IF shares a similar perturbation paradigm as DARTS-PT, while without any fine-tuning. Rather than using the validation performance drop to indicate the operation importance, DARTS-IF leverages influence functions to predict the loss change. Different from DARTS-IF and DARTS-PT, our DARTS-IM has a similar paradigm as DARTS, while whose operation importance is calculated based on $\mathcal{I_M}$, rather that the optimized $\alpha$. In addition, the operation importance in our DARTS-IM can be calculated by one shot, rather than iteratively going through all candidate operations in a supernet as the perturbation paradigm.
>
> 2. Lacking qualitative analysis on skip-connection and tabular analysis compared to DARTS-PT, especially on S1-S4.
>
> Thanks for the constructive comment. We compared our proposed DARTS-IM with the DARTS for the test errors in Table 1 and Table 3 on NAS-Bench-201 and DARTS space, respectively. We qualitatively compared the number of skip-connection on NAS-Bench-201 in Sec. 6.1. **“Although DARTS-PT can partially relieve this issue by selecting one or two more convolutional operations, it still prefers skip-connections (selects 3 skip-connections for 6 edges) where directly removing them still significantly affect the performance. On the contrary, our DARTS-IF and DARTS-IM can find valid architectures with more convolutional operations (with only one skip-connection).”**
>
> In addition, we also discussed the searched architectures on space S1-S4 in the Appendix G, where our DARTS-IM selects fewer skip connections compared with DARTS and DARTS-PT, especially on S1 and S2.
>
> Following the suggestion, **we add new comparison experiments on NAS-Bench-1shot1 with DARTS-PT in Table2, and also add a new Table 7** in the Appendix to summarize the test errors of different DARTS baselines on S1-S4, where our DARTS-IM achieves the best results in most cases.
>
>
> 3. In Section 5, instead of using the Neuman-series to compute the inverse Hessian-vector product, the authors proposed to use empirical Fisher and compute the term using the Sherman-Morrison formula. I believe that the manuscript lacks a proper explanation for this approximation.
>
> Thanks for the constructive suggestion, and we added the motivation and explanation for this approximation in the revision. As we know, the true Fisher information matrix converges to the Hessian matrix with the training loss approaches to zero. However, it is usually infeasible to calculate the exact Hessian or Fisher matrix in a large neural network. A practical approximation is called as empirical Fisher, which has been verified to be effective in a variety of applications [1, 2]. In addition, an important advantage of empirical Fisher is that it allows approximating the inverse of the Hessian based on the Sherman-Morrison formula. Although the empirical Fisher can not exactly match the true Fisher and there are several active researches on its applicability [3, 5, 6], the empirical Fisher is significantly more computationally efficient, and obtains a good trade-off between the approximation and practical efficiency [4].
>
> 4. Additional hyperparameters setting description.
>
> The supernet training part in our DARTS-IM is exactly the same as DARTS; the only difference is from the operation selection part. In calculating the influence magnitude in our DARTS-IM, we have three more hyperparameters, $\gamma$ for Neumann series approximation, $ \eta$ for Sherman-Morrison approximation, and batch size $N$ for both. In our experiments, we set $\gamma$ the same as the learning rate and $ \eta$ is the weight decay for regularization by default, which can be obtained from the optimizer. We also conduct hyperparameter analysis on N in Figure 2. We added these hyperparameter setting descriptions in the algorithm framework description in Appendix H.
>
>
> [1] Natural gradient works efficiently in learning. Neural Computation, 1998
>
> [2] Second order derivatives for network pruning: Optimal brain surgeon, NIPS 1992
>
> [3] Limitations of the empirical fisher approximation for natural gradient descent, NeurIPS 2019
>
> [4] WoodFisher: Efficient Second-Order Approximation for Neural Network Compression, NeurIPS 2020
>
> [5] M-FAC: Efficient Matrix-Free Approximations of, Second-Order Information, NeurIPS 2021
>
> [6] On the interplay between noise and curvature and its effect on optimization and generalization

---

> > ### Author Response · Authors · 2022-08-02
> > **Response (2/2)**
> >
> > 5. Equation 3, H−1∇x∇θL(z,θ∗) should be replaced with H−1∇θL(z,θ∗).
> >
> > Thanks for point out. This is a typo, and we have revised it accordingly.
> >
> > 6.The gradient of the loss equals 0 in equation 6.
> >
> > Thanks for the suggestion.  We add this general assumption in the main text, which can be also found the Appendix in the derivation of Equation 6.
> >
> > 7.Is there a reason why NS performs poorly compared to SH?
> >
> > We pointed out the drawbacks of Neumann series in Sec. 5. For example, NS makes several hard assumptions, e.g., $\mathcal{L}$ is $\lambda$-strongly convex, and it is not easy to select an appropriate $\gamma$. The work \cite{basu2020influence} empirically showed that the stochastic Neumann series estimation for IHVP is somehow erroneous, especially when the network is deep. The paper [7] leveraged NS for the hypergradient approximation in DARTS, and the author empirically found that $K=2$ is large enough to approximate IHVPs while with less computational cost. In this paper, we also observed a similar phenomenon and also set $K=2$ by default in our NS. We give a discussion on K in our revision in the Appendix B1.
> >
> > 8.Is there a reason why DARTS-IF performs worse than DARTS-IM in Table 1?
> >
> > The main difference between DARTS-IF and DARTS-IM is the change on $\alpha$ when leveraging influence functions to approximate the loss change. In our DARTS-IF, we consider to remove a single operation from candidate operations in one edge, while DARTS-IM only apply an infinitesimal change on $\alpha$. In our Corollary 1, we show that the error bound of the approximated loss increase with the change of $\alpha$, which motivates DARTS-IM to applies an infinitesimal change on $\alpha$ and design the influence magnitude accordingly. DARTS-IF and DARTS-IM both can use SM and NS for for iHVP approximation.
> >
> > 9.Should be described finite-difference methods to compute the matrix-vector product in the main text.
> >
> > Thanks for the constructive suggestion. We revised accordingly and add the description in the maintext.
> >
> > 10.The results of PT-DARTS in Table 3 lower than the previously published baseline.
> >
> > Yes, the best run of DARTS-PT is 2.48. However, in the second column of Table 3 ("Single" column which means a single architecture with multi-run rather than the best run), we report the mean of the best searched architecture under several runs, whose numbers are from the original papers. The third column reports the statistical result of architectures after three searches. We revise Table 3 and report the statistical results for the best searched architecture in the “Single” column.
> >
> > 11. What Figure 4 is doing. Why does DARTS fail after 20 epochs of training? Using different colours for DARTS and SM and the comparison with PT-DARTS would be helpful.
> >
> > YES, it performs operation selection based on the supernet after x epochs supernet training. DARTS fails after 20 due to the skip connection which is also observed by DARTS-PT. Thanks for the suggestion, and we have redrawn the figures accordingly.
> >
> > [7] M. Zhang, S. Su, S. Pan, X. Chang, E. Abbasnejad, and R. Haffari. idarts: Differentiable architecture search with stochastic implicit gradients. ICML, 2021

---

> > > ### Comment · Reviewer_D82N · 2022-08-04
> > > **Reviewer D82N Response**
> > >
> > > I thank the authors for the detailed rebuttal and I acknowledge that I have read other reviewers' comments and the author's reply. Please see the below responses.
> > >
> > > > We add a new section in Appendix H to describe DARTS-IF and DARTS-IM (which are sketched in Algorithm 1 and Algorithm 2, respectively) and the hyperparameters setting in these two algorithms.
> > >
> > > Thank you for the revision. I believe that these will help the readers understand the idea more clearly. However, there seem to be several issues with the Algorithm. For example, edge $e$ was never removed according to Algorithm 1 but is restored in line 9. Moreover, both algorithms 1 and 2 do not describe additional hyperparameters for iHVP. I believe these algorithms are one of the essential diagrams in the paper and must be accurate and detailed.
> > >
> > > > Lacking qualitative analysis on skip-connection and tabular analysis compared to DARTS-PT, especially on S1-S4.
> > >
> > > I would like to note that one of the points from the weakness section was on lack of quantitative analysis (not the lack of qualitative analysis). Thank you for pointing out more experimental results. It would be helpful if the author made the best run in bold in Table 7 (which also applies to other tables). In Table 7, DARTS-PT outperforms DARTS-IM for some search spaces (e.g., C10-S3, C10-S4). Do the authors have any hypothesis on why this is the case?
> > >
> > > >  As we know, the true Fisher information matrix converges to the Hessian matrix with the training loss approaches to zero. However, it is usually infeasible to calculate the exact Hessian or Fisher matrix in a large neural network.
> > >
> > > Thank you for adding these comments to the paper. As you mentioned, while true Fisher converges to the Hessian when the training loss is 0, it is not the case for the empirical Fisher. I am still concerned about the validity of various approximations made together in the paper. The second-order gradient is computed using the finite difference (which is very crude), and the Hessian is approximated by empirical Fisher.
> > >
> > > > Is there a reason why NS performs poorly compared to SH?
> > >
> > > I agree with the authors that NS requires additional hyperparameter (scaling) tuning. Another question I had was if the authors tried tuning the damping in NS.
> > >
> > > Moreover, I encourage the authors to update incorrectly scaled figures (e.g., Figure 3 and Figure 7).
> > >
> > > Overall, I agree with reviewer 3tjK that the paper makes a solid attempt to improve operation selection in DARTS. However, I believe there needs to be a more careful study of the accuracy of these approximations (e.g., finite difference, empirical Fisher). While the paper makes some improvements over PT-DARTS (although it fails sometimes as shown in Table 7), I am still not convinced that the algorithm uses advantages from second-order approximation. I have increased my score as some of my concerns have been addressed by the authors, but I still believe there could be significant improvements in the analysis and experiments.

---

> > > > ### Author Response · Authors · 2022-08-07
> > > > **Response to the remaining concerns**
> > > >
> > > > Thank you for your constructive comments! We are pleased to follow your suggestion to improve our paper. All new revisions are marked with blue. We hope our answers below address your concerns, especially on the second-order information approximation.
> > > >
> > > > 1. More detailed description of Algorithm 1 and 2.
> > > >
> > > > Thanks for this comment. We would like to give more details on our Algorithms, including explaining the flow of the algorithm, the hyperparameter settings, and the difference between DARTS-PT, DARTS-IF, and DARTS-IM. We re-organized our Algorithm 1, where line 9 is removed as DARTS-IF never removes an edge from the supernet.
> > > >
> > > > 2. In Table 7, DARTS-PT outperforms DARTS-IM for some search spaces (e.g., C10-S3, C10-S4). Do the authors have any hypothesis on why this is the case?
> > > >
> > > > We discussed the potential reasons in several situations that DARTS-PT outperforms DARTS-IM from line 780. Apart from some search space properties in S3 and S4, we also point out an advantage in DARTS-PT, which only locally compares the operations' strength in each edge and a fine-tuning is conducted after every edge discretization. Although this perturbation strategy is more time-consuming, it can reduce the negative operation coupling in our DARTS-IM, which globally compares all operations in a supernet. To further improve our DARTS-IM, a simple and straight approach is to introduce the perturbation strategy to our DARTS-IM, which we leave for our future work.
> > > >
> > > > 3. **Concerns on the second-order gradient is computed using the finite difference (which is very crude), and the Hessian is approximated by empirical Fisher.**
> > > >
> > > > The finite difference in our DARTS-IM inherits from the DARTS which also uses it to calculate $ \frac{\partial^2 \mathcal{L}(\theta, \alpha)}{\partial \theta \partial \alpha}$, so as efficiently calculate the second-order information. In addition, DARTS, which uses the one-step unrolling, simply considers the Hessian as an identity matrix with crude approximation [1]. The empirical successes from DARTS partially verify the practical implementation for these crude approximations.
> > > >
> > > > We understand it also brings error when we consider an empirical fisher to approximate Hessian. So, we conduct an ablation study to empirically analyze several common approaches for the Hessian approximation. We consider the **identity approximation** (DARTS-IM-I), **diagonal approximation** (DARTS-IM-D), the **Neumann series approximation** (DARTS-IM-NS), and our **Sherman-Morrison approximation** (DARTS-IM-SM), and the comparison experimental results can be found in Table 6. Partial experiments can be found in the following table. We can find that our Sherman-Morrison approximation achieves the best results.
> > > >
> > > > |Dataset|CIFAR-10(valid)|CIFAR-10(test)|CIFAR-100(valid)|CIFAR-100 (test)|ImageNet-16-120(valid)|ImageNet-16-120(test)|
> > > > |---|----|----|----|----|----|----|
> > > > | DARTS-IM-I| 89.89±0.24| 93.11±0.17 | 69.50±0.60| 70.17±0.60|44.45±0.73| 44.44±0.32|
> > > > |DARTS-IM-D|89.62±1.70| 92.82±1.20| 69.17±2.61| 69.45±2.72|42.16±3.97 |42.07±3.50|
> > > > |DARTS-IM-NS |90.05±0.51|  93.35±0.38|  69.96±1.13|  70.26±1.13|  44.43±0.95|  44.03±0.75|
> > > > |DARTS-IM-SM|**90.92±0.34** |**93.61±0.23** |**71.21±0.55** |**71.31±0.40** |**44.70±0.74** |**44.98±0.36**|
> > > > | |
> > > >
> > > >
> > > > 4. If the authors tried tuning the damping in NS.
> > > >
> > > > Different from SM, NS has a key hyperparameter $\gamma$ to be tuned which is assumed to be small enough that $\gamma<\frac{1}{\sigma_{max}}$. In this way, we conduct the hyperparameter analysis on $\gamma$ for our DARTS-IM when employing the Neumann series approximation. A **new Figure 6** compares the performance of DARTS-IM with different $\gamma$ on the NAS-Bench-201. As shown, the Neumann series approximation is sensitive to the $\gamma$, where a large $\gamma$ significantly deteriorates the performance of DARTS-IM. For example, a $\gamma=0.05$ even makes our DART-IM similar to the performance of the random baseline. Compared with the Neumann series, we found that the Sherman-Morrison approximation is more robust.
> > > >
> > > > [1] M. Zhang, S. Su, S. Pan, X. Chang, E. Abbasnejad, and R. Haffari. idarts: Differentiable architecture search with stochastic implicit gradients. ICML, 2021

---

> > > > > ### Comment · Reviewer_D82N · 2022-08-08
> > > > > **Reviewer D82N Response**
> > > > >
> > > > > Thank you again for your detailed response!
> > > > >
> > > > > > More detailed description of Algorithm 1 and 2.
> > > > >
> > > > > Thank you for updating the algorithm. I confirm that my concerns regarding clarity have been resolved.
> > > > >
> > > > > > In Table 7, DARTS-PT outperforms DARTS-IM for some search spaces (e.g., C10-S3, C10-S4). Do the authors have any hypothesis on why this is the case?
> > > > >
> > > > > I don't believe the authors' hypothesis is persuasive (and well supported). However, I agree that gradual discretization would further help the method and should be explored in the future.
> > > > >
> > > > > > Concerns on the second-order gradient are computed using the finite difference (which is very crude), and the Hessian is approximated by empirical Fisher.
> > > > >
> > > > > I understand that the DARTS uses the finite difference and one-step unrolling to compute the gradient. The response gradient (in gradient-based bilevel optimization) can be roughly framed as computing the influence score on the operations. However, to my knowledge, the NAS community is still unsure if this influence score is helpful in DARTS, and many papers simply omit this term when updating the architecture operation.
> > > > >
> > > > > The experiments (in Table 6) show that DARTS-IM-NS and DARTS-IM-SM achieve better performance by finding a better-performing cell in this particular search space. Still, this does not necessarily mean that it benefits from second-order information. To make my logic more precise, better performance in this evaluation scheme does not imply the algorithm successfully utilizes second-order information. A simple experiment that will make the paper more convincing is to compute CG on a supernet with maybe one cell and examine how accurate NS and SM are (including exact calculation of the second-order mixed gradient instead of finite difference). Here, although SM may make a more inaccurate approximation than NS, some inductive bias may be leading to an increase in the validation accuracy. Of course, this speculation may be incorrect, but the paper does not include any discussion on this. I don't expect the authors to conduct such an experiment in this review period, but I believe this would make the paper convincing.
> > > > >
> > > > > Another interesting observation in Table 6 is that the diagonal approximation performs worse than identity.
> > > > >
> > > > > > If the authors tried tuning the damping in NS.
> > > > >
> > > > > Thank you for the clarification and for including a new figure. It is interesting that using larger damping lead to performance similar to a random baseline. I thought it would converge to something similar to identity-diagonal approximation.
> > > > >
> > > > > Assuming that the authors will modify some incorrectly scaled figures in the next revision, my concerns about clarity have been resolved. However, my concerns on justification and empirical analysis of second-order information still remain. I updated my score to reflect my thoughts.

---

> > > > > > ### Author Response · Authors · 2022-08-09
> > > > > > **Thanks for the constructive suggestion.**
> > > > > >
> > > > > > I really appreciate your constructive comments, which help improve our paper to a large extent.

---

### Official Review · Reviewer_pKq9 · 2022-07-27

**Rating:** 3
**Confidence:** 5
**Soundness:** 3 good
**Presentation:** 3 good
**Contribution:** 2 fair

**Summary:**

The assumption original DARTS is lack of theoretical guarantees. This paper supports the theoretical explanation of operation selection in DARTS. It shows that the operation weights is not only associated to the magnitude but also a second-order information called Influential Magnitude. By using this influential magnitude, DARTS algorithms performed better than other baselines on NASBench201, NAS-Bench-1shot1, Cifar10 and ImageNet search space.


**Questions:**

1. Why not compare DARTS-PT in NAS-Bench-1shot1 to make it consistent with other experiments?
2. How many runs in table2 and table3?
3. Is the Test error Multi includes singe test in the first column of table3? If so, why the 2.50% for DARTS-IM not in the range of 2.70±0.18? If not, why not merge them?


**Ethics Review Area:**

["I don’t know"]

**Limitations:**

See weaknesses.


**Strengths And Weaknesses:**

+ Paper is clear and easy to follow.
+ Give a concretely theoretically interpretation for operation selection in DARTS, which is one of the most popular search algorithm in NAS community.
+ Search performance by using proposed operation selection method is improved on different search spaces compared to DARTS and its variants .
+ The proposed method is able to get the stable results in multiple search runs.
+ Code is available.

- The practical implementation should be based on many assumptions and approximations.
- The improvement of ImageNet and Cifar10 is marginal.
- The words/characters in Figure3 is very unclear and hard to understand.
- Not too much practical.

---

> ### Author Response · Authors · 2022-08-02
> **Response**
>
> Thanks for the constructive comments! We hope our answers below address all your concerns.
>
> 1. The practical implementation should be based on many assumptions and approximations.
>
> In this paper, we consider Assumption1-3, which are common in analyzing the convergence of bi-level optimization [1,2,3],  to derive Corollary 1. We should notice that, Corollary 1 is to show that a large change on $\alpha$ will incur a large error bound, so as motivates our DARTS-IM to consider an infinitesimal change on $\alpha$. The practical implementation of our DART-IM does not dependent on these three assumptions. In the adopted IHVPs for practical implementation, we also only consider the commonly used assumptions for approximations. The other assumption is that we follow [4] that  the third and higher derivatives of the loss function $\mathcal{L}$ at optimum is zero or sufficiently small.
>
> In addition, in the practical implementation, we only have three more hyperparameters, $\gamma$ for Neumann series approximation, $ \eta$ for Sherman-Morrison approximation, and batch size $N$ for both. In our experiments, we set $\gamma$ same as the learning rate and $ \eta$ is the weight decay for regularization by default, which both can be obtained from the optimizer. Thus, our method is very easy to be implemented, and extensive experiments also verify the effectiveness of the proposed method.
>
> In addition, our work is to motivate more people pay attention on the operation selection as we found that our method can greatly improve the performance of DARTS, even we consider several assumption and approximation. This paper verified that operation selection, compared with supernet training, is a promising direction in the NAS community.
>
> 2. The improvement of ImageNet and Cifar10 is marginal.
>
> As mentioned, this paper is mainly focusing on the operation selection part in differentiable architecture search. Compared with two baseline, DARTS and DART-PT which use the \textit{argmax} and perturbation for operation selection respectively, our DARTS-IM obtained considerable improvement with 25.0/7.6  top1/top5 test error for ImageNet on DARTS space (comparing DARTS, 26.9 / 8.7, and DARTS-PT, 26.1 / 8.2). On the CIFAR10 for DARTS space, our DART-IM also obtains a performance improvement, 2.50 $\pm$0.10 comparing DARTS (2.76$\pm$0.09) and DARTS-PT (2.61$\pm$0.08). In the NAS-Bench-201 space, we can also see a considerale performance increase from our DARTS-PT. For example, our DART-IM achieves 93.61$\pm$0.23, 71.31$\pm$0.40, 44.98$\pm$0.36 accuracy on the CIFAR10, CIFAR100, and ImageNet, respectively, significantly outperforming DART (54.30$\pm$0.00, 15.61$\pm$0.00, 16.32$\pm$0.00) and DARTS-PT (89.63$\pm$0.19,62.35$\pm$2.14, 36.51$\pm$2.13).
>
> Rather than focusing on the operation selection, most existing works focus on the other orthogonal part, supernet training in differentiable architecture search. Taking the PC-DARTS as a competitive alternative, it can also improve the DARTS baseline by large margins. Compared to the supernet training, much less works focus on the operation selection, and our work is to motivate more people pay attention to this part as we found that our method can also greatly improve the performance of DARTS, with even more improvements than those works focusing on the supernet training.
>
> 3. The words/characters in Figure3 is very unclear and hard to understand.
>
> Thanks for the suggestion. We redraw these figures for better visualization.
>
> 4. Why not compare DARTS-PT in NAS-Bench-1shot1 to make it consistent with other experiments?
>
> Thanks for the constructive comment. The original paper of DARTS-PT did not provide the results on NAS-Bench-1shot1. **To make it consistent, we follow the suggestion to reproduce the results of DARTS-PT on NAS-Bench-1shot1, and revise the Table 2 accordingly.**
>
> 5. How many runs in table2 and table3?
>
> In our experiments, we run the experiments with 3 times.
>
> 6. Is the Test error Multi includes singe test in the first column of table3? If so, why the 2.50% for DARTS-IM not in the range of 2.70±0.18? If not, why not merge them?
>
> No, the “Single” means that we only run the best searched architecture for several time and report the mean test error. In the “Multi”, we search architectures for several times and reports the statistical results of several searched architectures. We **revise Table 3 and report the statistical results** for the best searched architecture in the “Single” column for consistency.
>
> [1] S. Ghadimi and M. Wang. Approximation methods for bilevel programming. 2018
>
> [2] R. Grazzi, M. Pontil, and S. Salzo. Convergence properties of stochastic hypergradients, AISTATS 2021
>
> [3] M. Zhang, S. Su, S. Pan, X. Chang, E. Abbasnejad, and R. Haffari. idarts: Differentiable architecture search with stochastic implicit gradients. ICML, 2021
>
> [4] S. Basu, X. You, and S. Feizi. On second-order group influence functions for black-box predictions. ICML 2020

---

> ### Author Response · Authors · 2022-08-08
> **We would love to hear your feedback on our rebuttal**
>
> Dear Reviewer pKq9 ,
>
> As the discussion period is close to the end and we have not yet heard back from you, we wanted to reach out to see if our rebuttal response has addressed your concerns.
>
> We are more than happy to discuss further if you have any further concerns and issues, please kindly let us know your feedback. Thank you for your time and help!

---

### Author Response · Authors · 2022-08-07
**Paper updated and further discussion**

Dear reviewers,

We have uploaded a paper revision addressing your concerns and suggestions, and making other improvements:

1. Added algorithm frameworks for DARTS-IF and DARTS-IF, along with description and hyperparameter setting.

2. Described assumptions on the local optimal and described the finite difference approximation in the main text.

3. Add the motivation to use empirical Fisher based Sherman-Morrison approximation.

4. Changed Figure 4 and added the plots of DARTS-PT.

5. Added discussion in the hyperparameter setting for DART-IM with Neumann series. Conducted hyperparameter analysis on $\gamma$ and added Figure 6.

6. Added the ablation study on the different Hessian approximations for DARTS-IM.

7. Added tabular results on the S1-S4 search space.

8. Added comparison results with DARTS-PT on NAS-Bench-1shot1.

All changes are marked with red or blue color.

We appreciate all reviewers for the hard work and helpful comments. We would like to address all reviewers’ concerns in the corresponding responses.

---

### Meta-Review · Area_Chair_ZfLM · 2022-08-28

**Recommendation:** Accept
**Confidence:** Certain

**Metareview:**

The paper proposes a novel architecture selection method for differentiable NAS based on influence function calculation. During the discussions reviewers still found several weaknesses of the paper, including 1) the novelty of the application of influence function in NAS, and it requires a series of approximation 2) The updated results from the authors show that the proposed method may not outperform existing approaches (DARTS-PT) in some cases.

However, since architecture selection is an important but under-studied problem in NAS, and the proposed method is well-motivated, we still think this paper is a solid and novel attempt on improving NAS. We therefore decide to accept the paper.

**Award:**

No

---

### Decision · Program_Chairs · 2022-09-14

Accept